# A Survey of Fear for Others, Fear for Self, and Pandemic Anxiety Predicting Intention to Take the First Booster Vaccine to Combat COVID-19

**DOI:** 10.3390/vaccines12010047

**Published:** 2023-12-31

**Authors:** Joseph N. Scudder, Dennis P. DeBeck

**Affiliations:** 1Department of Communication, Northern Illinois University, DeKalb, IL 60115, USA; 2Department of Communication, University of South Florida, Tampa, FL 33620, USA; ddebeck@usf.edu

**Keywords:** COVID-19 first booster vaccine, fear for others, fear for self, COVID-19 anxiety, COVID-19 vaccination hesitancy

## Abstract

This study examined the impact of fear and anxiety on the intent to take the first COVID-19 booster vaccine. The objective of this study is to provide guidance for messaging campaigns of public health practitioners. A survey approach provided insights about individuals’ emotions of fear and anxiety related to adopting the first booster vaccine for the Coronavirus disease 2019 (COVID-19). Methods: Three independent variables were considered in their ability to predict the intent to take the first COVID-19 booster vaccine (BINT): Fear for Others (FOTH), Fear for SELF (FSELF), and COVID-19 Anxiety (CANX). Results: The confirmatory factor analysis supported an underlying three-factor solution for three central emotions in this study. A path analysis indicated significant direct effects for FOTH and FSELF in the prediction of BINT. The interdependent nature of these variables on the intent to get the first booster vaccine also was indicated by significant indirect effects. Discussion: Fear should be more precisely refined to include the fear for others (FOTH) beyond consideration of the fear for self (FSELF) from the impact of COVID-19. Conclusions: FOTH and FSELF were demonstrated to be direct predictors of BINT. CANX was only found to be significant as part of indirect effects impacting BINT. Future investigation should be given to the mediating role of anxiety with FOTH and FSELF as the context changes.

## 1. Introduction

Fear is directly related to the behavioral intention to be vaccinated for the coronavirus disease 2019 (COVID-19) [1,2,3,4]. Fear and anxiety related to the intent to take the first COVID-19 booster vaccine were the focus of this study, for the purpose of providing public health practitioners better guidance for their messaging campaigns. Fear of COVID-19, as viewed through the lens of productive risk management, is a positive outcome when it is not countered by excessive anxiety that can lead to hesitation [2]. Although many other factors impacted the adoption of the first COVID-19 booster vaccine, the focus here is reconsideration of traditional approaches in vaccination campaigns that placed the individual as the sole focus of health threats. It is proposed that communicable diseases like COVID-19 require focus on fear for others and anxiety in relational contexts, in addition to the traditional focus on fear for the self.

### 1.1. Challenges Facing First Responders (FR) and Frontline Healthcare Professionals (FHP)

The rollout of the first booster vaccine benefited from what was learned from the emotional responses of first responders (FR) and frontline healthcare professionals (FHP) during the initial days of the COVID-19 pandemic. Those on the front line fighting the pandemic faced multiple types of fear and anxiety [5,6,7]. Many faced the fear and anxiety of those they were helping—especially those headed toward intensive-care units with life-threatening conditions. 

Crowe et al. [5] provide striking commentary from the experiences of several nurses who were particularly stressed by working conditions while entertaining the possibilities of bringing the disease home to their loved ones. One nurse said, “My first emotion was fear. I feared for my family. I was afraid working on the frontline would put my family at risk. I really didn’t want to come to work” (p. 6). Another nurse commented, “I worry about getting sick from COVID or worse—spreading it to a loved one or to a vulnerable person and feeling guilty about this” (p. 6). McAlearney et al. [6] provided similar testimony from one police officer who stated, “I was worried [about] taking it home to my family members like some of the more high-risk people in my family or the baby at home” (p. 7). Other HC providers, such as dentists with close contacts with their patients, reported fear of contracting the disease from patients and carrying it home to their families [8]. Such examples provided direction for the rollout of the first booster vaccine.

These examples of experienced threats from FP and HCP did not fit the descriptions of fear and anxiety that are articulated in many public health campaigns and most experimental research regarding fear appeals. Much academic research regarding fear has failed to explicate its complexity, because it has often been considered in controlled conditions such as experimental studies of fear appeals (see the meta-analyses of Bigsby and Albarracín [9], Tannenbaum et al. [10] and Witte and Allen [11]. However, experimental studies need to control other confounding variables as much as possible. While valuable in isolating the impact of influence of using fear appeals, this substantial body of research studies in these meta-analyses regarding fear appeals does not address the multiple types of fear and anxiety that FHP and FR faced.

### 1.2. Second-Wave Threat of COVID-19

The opportunity to examine the influence of the FOTH in comparison to FSELF and CANX came as the Delta variant emerged in June 2021 in the USA in a second wave of COVID-19 [12], as well as worldwide [13]. The severe impact of the Delta variant became evident, despite the original vaccine being widely available. The Delta variant was estimated to be far more virulent than the virus in the original epidemic. It increased transmissibility and decreased vaccine efficacy [14,15]. After the initial peak in hospitalizations and deaths due to COVID-19 subsided in the USA, the death rate skyrocketed ten-fold from July to September 2021 and did not move below 5000 deaths per week until the end of March 2022.

### 1.3. Experimental Fear Appeal Research Related to COVID-19

Recent experimental considerations of fear in the COVID-19 pandemic by Roberto and associates [16] provided practical insights into intentions to get the COVID-19 vaccines, using Witte’s [17] extended parallel processing model (EPPM). Roberto et al.’s earlier experimental research applied the EPPM to receiving the influenza vaccine [18]. These two studies contributed substantial insights into the use of fear appeals in recommended vaccine usage that extend beyond emotional components to other variables.

### 1.4. Altruistic Fear (FOTH)

The discussion of FOTH in health contexts is not a new topic, but it has been long neglected. It was raised early in public health research by Hochbaum [19] in 1958, when he was encouraging people to participate in X-ray screenings for tuberculosis. He recognized that fear for others existed as he attempted to persuade adults to have preventative X-ray screenings for tuberculosis. Although Hochbaum’s early research led to the development of the influential health belief model (HBM), his more nuanced perspective on health-related fear extending beyond the individual has largely been ignored by those using the HBM framework. However, FOTH has appeared in studies of crime and safety in the field of sociology for the past three decades.

Sociologist Mark Warr [20] drew attention in 1992 to the prevalence of FOTH that he labelled “altruistic fear” (p. 723). Warr and Ellison [21] provided an expanded treatment of the topic in their summary:

Research on fear of crime in the United States has concentrated on personal fear while overlooking the fear that people have for others in their lives—children, spouses, friends—whose safety they value…. Altruistic fear (fear for others) has a distinctive structure in family households and is more common and often more intense than personal fear. (p. 551)

They further amplify these sentiments saying that “parents may undertake more extreme or determined measures to protect their offspring than they would to protect themselves” (p. 552).

Roberto and Zhou [16] suggested that growing evidence supports the proposition that fear for others can influence the health intentions of behaviors of others. Chen and Chen [22] found that fear influenced smoking cessation. The few COVID-19 studies that do measure FOTH from the negative impact of the disease usually also address fear for the negative impact on oneself. Cori et al. [23] provided evidence from an Italian sample, showing that fear for the family of infection from COVID-19 was greater than FSELF. This was particularly a concern for health care workers fearing bringing COVID-19 home to their families [24]. Nurses often experienced fear of infecting others and stress that was exacerbated by the lack of personal protective equipment (PPE) in many locations [25].

### 1.5. Distinct Nature of Anxiety

An important part of this study is the recognition that FOTH is distinct from anxiety. This distinction regarding COVID-19 vaccination intention was clear in the research of Scrima et al. [2]. CANX in our study refers to state anxiety rather than trait anxiety. We recognize this is contested ground. Ahorsu et al. [26] combined fear and anxiety items in their Fear of COVID-19 Scale, which has been used in a multitude of studies. Witte [17] included an anxiety item in her measure of fear in her EPPM model, as did the recent test of that model, in the context of COVID-19, by Roberto and Zhou [16]. In contrast, we side with researchers who make the distinction of fear and anxiety as being separate processes [27,28]. So et al. [29] suggested that anxiety was a separate emotion, in addition to fear, that could emerge in response to a threat. In an experimental test of their perspective, they found support for considering fear and anxiety as separate constructs, in an information-seeking experiment regarding the meningococcal vaccination. Their study advanced understanding of the relationship between fear and anxiety by demonstrating that anxiety may be the better predictor of health-related outcomes in some contexts, but the reasons why were not apparent.

Heeren [30] argued that the conceptual and functional distinction between fear and anxiety could further advance Schimmenti et al.’s [31] framework regarding fear in the COVID-19 pandemic. Heeren [30] suggested that fear is grounded in the present and is a transient state in response to a specific threat like COVID-19. In contrast, he argued that anxiety is a future-oriented and long-lasting response to an unclear and unpredictable future. This perspective emphasizes that anxiety responds to uncertainty.

Ohman [32] agreed with Heeren [30] that “fear and anxiety are different emotion” (p. 123). However, he added a new insight into the difference between fear and anxiety, suggesting that fear happens when a person is actively coping with a threat, but anxiety occurs when that person is not effectively able to resolve or fully cope with that threat. The slight difference in his perspective is useful, because it acknowledges that fear may not end with a complete resolution of the problem and anxiety deals with an ongoing threat; such is the case with the original COVID-19 vaccine and subsequent shots not eliminating the possibility of contracting the disease.

More specific to the COVID-19 context, several studies specifically considered “worry”, which is usually associated with anxiety, without specifying any type of fear. Beyond parents being worried they would bring COVID-19 home to their families, they also worried that their children would contract it at school [33]. Similarly, Barzilay et al. [34] found that participants were more worried about a family member contracting or unintentionally infecting others than getting it themselves. Asymptomatic individuals were a complicating factor, increasing stress for those concerned about infecting other family members.

## 2. Materials and Methods

Many unresolved questions remain about the relationships of FSELF, FOTH and CANX as independent measures in the prediction of the intent to get the booster vaccine. Although some evidence has been presented that FOTH has been important for first responders and frontline healthcare workers, those reports have not been systematically tested against reported fear for self and anxiety about COVID-19.

### 2.1. Research Questions

Past research provides little guidance regarding whether these measures are independent and predict unique variance with the intent to get a booster vaccine. Items in each of the scales used were answered in the context of COVID-19, rather than as generalized variables forming enduring traits. The first research question attempts to provide initial guidance about these relationships.

RQ_1_: Are FSELF, FOTH and CANX unique independent measures?

Based on the prior review, qualitative reports indicate that many individuals experience more FOTH of COVID-19 for significant others contracting it than FSELF, but research provides little indication of whether to expect FOTH to be a better predictor of vaccination intent than FSELF or CANX. On the other hand, some live in more individualistic and/or hedonistic environments, where self-interest is the focus guiding behavior. Others realize that they must survive for the well-being of their families. However, So et al. [29] highlighted the superiority of anxiety to fear in their work on meningitis prevention via vaccines. This raises issues about the relationship of the independent variable CANX to FOTH and FSELF, in regard to the dependent variable BINT. The experimental conditions of So et al. did not produce high levels of fear. Under more severe threat conditions in the actual life context of COVID-19, incomplete resolutions of fear could result, because no vaccine is 100% effectivebecause of uncertainty regarding the success rate of the vaccines. Doubts about the complete effectiveness of the vaccines could lead to more lingering anxiety [33]. Yet, prior research provides no concrete guidance as to whether CANX would take precedence, as it did in So et al. [29]. Past research is unclear what level of experienced anxiety regarding COVID-19 should be experienced in relation to FOTH and FSELF, in regard to COVID-19 booster vaccination behavior. This leads to the second research question, regarding the relationship of ANX to FOTH and FSELF, for the dependent variable BINT.

RQ_2_: Which of FOTH, FSELF or CANX of COVID-19 are unique predictors of BINT?

Traditional regression analysis has the limitation of treating the central independent variables as parallel paths that do not interact. More recent approaches, such as those proposed by Hayes [35], permit the consideration of the indirect effects of variables on each other. Such approaches allow the consideration of variables that are co-acting with each other. RQ_3_ considers whether indirect effects are occurring in relation to the intent to take the first boost vaccine.

RQ_3_: Are significant indirect effects present among FOTH, FSELF or CANX as predictors of BINT?

### 2.2. Procedures

This project began with the two researchers of this study considering how public health campaigns might better communicate the importance of getting the first COVID-19 booster vaccine to combat a still-raging epidemic after the original vaccines had been introduced. Shortly before the first booster vaccines became available in fall 2021, discussions were conducted across a wide range of individuals, asking why they believed people decided to take or not take the original COVID-19 vaccines. Of particular interest were comments participants made about affective elements of fear for passing the disease to their families—particularly their elders. This led to the generation of a set of research questions laying the foundations for the results reported here.

We then presented this project to the Institutional Review Board of each of our institutions. After approval, participants were recruited from Amazon Mechanical Turk (AMT). Crowston [36]. describes AMT as a system for crowdsourcing work that has been used in many academic fields, to find participants and collect data. AMT connects workers using its database with researchers needing assistance in tasks such as completing research surveys. The data are made available in the Appendix A for researchers to analyze with their own software. Researchers contract with AMT workers for clearly stated compensation. All AMT workers have established identification numbers. AMT workers have ratings to guide researchers about their past performance.

In our study, participants were compensated USD2.50 upon completion of the survey. Data collection took place between the first week of December 2021 and the middle of January 2022, when booster vaccinations took a sharp downturn, known as an elbow in the data trend. The potential impact of the following covariates was assessed: the original vaccine they received, political affiliation, gender, education, age, participation in religious meetings, adverse side effects from the original COVID-19, regularity of routine medical attention, whether a participant reported having COVID-19 and the severity experienced from having the disease.

### 2.3. Participants

A major requirement for participation was that participants had received the initial COVID-19 vaccine, but not yet the booster vaccine. They were also required to be fluent in English and over the age of 18. Incomplete surveys without sufficient data for analysis were excluded. Those not meeting these requirements were dropped from the participant pool. One downside of AMT is that access can sometimes be gained using automated online programs posing as individuals to illegally get compensated. A set of challenges were developed in the survey to eliminate illegitimate submissions. Those rejected could appeal that they were legitimate workers. These measures resulted in 30 submissions being excluded from further analysis. See the Appendix A for further details on informed consent.

### 2.4. Measures

The BINT, FOTH, FSELF and CANX scales ranged from a low of one to a high of seven. The scale for fear for self of COVID-19 (FSELF) included three items, measuring fear of the COVID-19 virus as an indicator of the perceived severity or threat of the disease. These items included: “I fear COVID-19”, “I fear dying from COVID-19” and “I fear lasting health complications from COVID-19”. The alpha reliability of this three-item scale was 0.84.

The scale for fear for others of COVID-19 (FOTH) comprised three items: “COVID-19 makes me fear for the well-being of my family who do not live with me”, “COVID-19 makes me fear for the well-being of my friends” and “COVID-19 makes me fear for the well-being of my family I live with”. The alpha reliability of this three-item scale was 0.87.

The scale for anxiety about COVID-19 (CANX) was constituted by three items: “Anxiety about COVID-19 is constantly with me”, “I am generally more anxious because of the new COVID-19 variants now appearing” and “I am more on edge when I am in public compared to before COVID-19”. The alpha reliability of this scale was 0.81.

The booster intention scale (BINT) was the dependent variable. It comprised two items on seven-point scales, with an alpha reliability of 0.89. The first item was “I received the initial COVID-19 vaccine doses already and I intend to get the COVID-19 booster dose as soon as possible”. The other item was “I received the initial COVID-19 vaccine doses already and I intend to get the COVID-19 booster dose in the near future”. 

## 3. Results

Rosenstock, Strecher, and Becker [37] suggested that getting people to take public health issues seriously involves “sufficient motivation (or health concern) to make health issues salient or relevant” (p. 177). The overall mean for BINT was 5.27 (SE = 0.07). The scale ranged from a low of one to a high of seven. This moderately high mean, for intent to take the first booster vaccine, suggests that participants were motivated to address the COVID-19 threat. Only 10.6% of participants had a mean of 3.0 or lower, with only 16 persons having a score of 1.0, indicating an absolute intent not to get the booster vaccine. 

### 3.1. Participants

There were 481 participants remaining after the data quality checks were performed. These data integrity check measures resulted in 30 submissions being excluded from further analysis. Participants averaged 38.0 years of age. Participants were predominantly male (56.1%) and white (78%). Three persons identified as transgender. Most participants reported they had completed their bachelor’s degree or beyond (71%). The sample leaned Democratic (60.1%), with 18.9% identifying as Republican, 17.5% as Independents and 3.5% as other. The average participant completed the survey in 28.4 min, with the fastest completion being at 6.8 min. The median time taken was 21.2 min. Only 5% completed the survey in less than 10 min. This provides evidence that most took this survey without rushing.

### 3.2. Impact of Covariates and Overall Relationships of the Variables

The party affiliation of the participants was the only of the assessed covariates that had a significant impact on whether participants reported the intention to get the first booster vaccine. Those affiliating with the Democratic party had a significant positive relationship (*r* = 0.24, *p* < 0.01) with intending to take the booster vaccine, versus Republicans (*r* = −0.12, *p* < 0.05) and Independents (*r* = −0.14, *p* < 0.01). Political affiliation as a covariate is not a surprise, with the politicization that occurred regarding COVID-19 vaccines. Those taking vaccines from different vaccine manufacturers did not significantly differ on BINT, nor did those who previously contracted COVID-19. Gender, age and education also had no significant impact on BINT. Those maintaining regular medical visits did not have a greater likelihood of taking the booster.

### 3.3. Research Question One

The first research question considered whether FSELF, FOTH and CANX are unique measures. A confirmatory factor analysis (see Figure 1) was conducted, to validate the integrity of the three emotions: FOTH, FSELF and CANX. The three-factor solution was not an excellent fit as drawn (χ^2^ (24, N = 481) = 76.81, *p* < 0.001). However, Kline [38] suggested that the global fit indices indicated a moderate fit. The GFI was 0.96. The RMSEA was 0.07 and was within its confidence range of 0.06 to 0.09. The RMR was a borderline 0.08 and the χ^2^*/df* ratio was 3.20. Kline maintains that global fit is not the entire story and that residuals among items should be examined at the local level. No standardized residuals exceeded the 0.05 level of significance (*p* > 0.05) across the items on the three scales. All items shared variance with their respective scale above the recommended 0.50 that Kline suggests. Six pairs of correlated error terms were the issues preventing a close-fitting model. Five of those problem error terms resulted from a pair including the item, “I fear dying from COVID-19”. This was the most extreme fear item on the scale that was deemed essential to retain. When those errors were permitted to correlate, the chi-square value became nonsignificant (*p* > 0.12). There were no significant cross-loadings of items with other factors. This combined evidence supports FOTH, FSELF and CANX being three separate constructs.

RQ_2_ asked which of FOTH, FSELF or CANX of COVID-19 were better predictors of BINT. The zero-order correlations reported in Table 1 indicated that fear for others (FOTH), COVID-19 anxiety (CANX) and fear for self (FSELF) all had significant and substantial relationships with intent to take the first booster vaccine (BINT). Confidence intervals at the 0.05 level of significance were constructed from the means and standards deviations in Table 1 of FOTH, CANX and FSELF. The three confidence intervals indicated that the mean of 5.09 for FOTH was significantly greater than the mean of 4.81 for CANX and the mean of 4.72 for FSELF. The CANX mean was also significantly greater than the mean for FSELF. Using IBM SPSS version 28 [39], FOTH, FSELF and CANX were entered simultaneously as independent variables into a standard regression analysis with BINT as the dependent variable, while controlling for Democratic, Republican and Independent political affiliation. Democratic affiliation was a significant predictor of BINT, (β = 0.16, *p* < 0.001). FOTH was the best single predictor of BINT (β = 0.44, *p* < 0.001). FSELF was the second-best predictor of BINT (β = 0.19, *p* < 0.001). However, CANX was not a significant predictor of BINT after FOTH, FSELF and affiliation with the Democratic party were entered, (β = 0.001, *p* > 0.99). The Republican and Independent parties were not significant predictors. Although these regression results provide the big picture, the answer is more complicated. Linear regression does not give much insight into the way these three variables function together.

To explore RQ_3_, the Hayes [35] PROCESS version 4.1 procedure 6 was used (see Figure 2) to assess where indirect effects occurred, in addition to the direct effects. PROCESS refers to a statistical application developed by Andrew Hayes to conduct conditional process analysis as an extension of traditional multiple regression. FOTH (β = 44, *p* < 0.0001) and FSELF (β = 19, *p* < 0.002) both had significant direct relationships with BINT. The significant standardized indirect effect of X > M2 > Y was 0.08, *p* < 0.01. The standardized indirect effect of X > M1 > M2 > Y was 0.06, *p* < 0.01. Thus, CANX was only a significant indirect predictor of BINT. Controlling for identification as a Democrat had a substantial effect (β = 35, *p* < 0.001). Neither Republican nor Independent party affiliation were significant covariates. The total effect that included direct and indirect effects of X on Y was 0.62. So, the answer to RQ_3_ is that significant indirect effects were present in Figure 2, which go beyond what traditional multiple regression revealed about the data.

Taken together, these direct and indirect results reveal patterns of more complicated relationships among FOTH, FSELF and CANX, that were not revealed in the standard regression analysis. FOTH had the greatest direct impact on BINT, as well as being part of the two significant indirect effects. FSELF had the second largest direct impact on BINT and was also part of two significant indirect effects. CANX played a secondary role, as part of the three-way indirect impact on BINT.

## 4. Discussion

### 4.1. Limitations

#### 4.1.1. Model Specifications

A natural question to ask is how the results change if we switch the X variable in Figure 2 to begin with FSELF and change M1 to be FOTH. Overall, the total effect of X on Y is 0.62, as it is presented in Figure 1. It drops only slightly to 0.52 when FSELF and FOTH are reversed. The three-way indirect effect of X > M1 > M2 > Y would disappear and the Democratic party covariate would be only 0.16.

It appears that CANX should not be the X variable in the Hayes [35] PROCESS analysis, because it reduces the overall total effect of the model to 0.50 and would result in anxiety having no direct relationship with the intent to take the booster vaccine. All of its impact on BINT would be indirect effects. Of course, that is a possibility, but does not provide the best fitting model. We originally started with FOTH in the X position in the PROCESS procedure, because the initial standard regression gave FOTH the strongest weight when entered simultaneously with FSELF and CANX. Based upon the results of Scrima et al. [2], our suspicion is that CANX might play a direct and possibly the leading role if studying a vaccination hesitant population.

The zero-order correlations in Figure 1 also indicate that FOTH had the strongest relationship with BINT of *r* = 0.63, but FSELF also had a moderately high zero-order correlation with BINT, *r* = 0.57. Our conclusion is that both FOTH and FSELF are important predictors of BINT. FOTH holds an advantage in predicting BINT based on the results of this study. CANX was only a significant indirect predictor of booster intent in the model in Figure 2, but it had a zero-order correlation of 0.52 with BINT. None of those relationships on their own were low. Hayes and Rockwood [40] make clear that causal models using the PROCESS approach cannot be absolutely validated by statistics. They suggest that statistical methods can be used to quantify causal arguments, but these are not the only considerations in establishing cause–effect relationships. Moreover, they maintain that solid research design and strong theoretical arguments are needed as well. Given that causal relationships among FOTH, FSELF and CANX have not been extensively considered in past research, the model in Figure 2 needs more testing under different conditions to establish whether the relationships in Figure 2 extend beyond this sample and this context. Even now, the model may change toward more focus on CANX as the pandemic lessens or as the population studied has more vaccine-hesitant participants. The patterns of the significant indirect effects indicated in the results of this study must be clarified with further research, but they do provide evidence that the three variables are interconnected and mutually dependent.

#### 4.1.2. Sampling

Using AMT allowed us to have a distribution of responses across many states and regions, but it was not a random sample of those who had taken the original COVID-19 vaccine across gender, age, economic level, and the political spectrum. Therefore, it had the limitations of being a convenience sample. A much larger sample may address some of the issues of restricted variable ranges discussed in the next section.

### 4.2. Restricted Variable Ranges

There were not many participants who had a low intention to take the first booster vaccine. Only 13.5% reported an intention to take the booster vaccine under the midpoint of the scale of 3.50. Similarly, there were very few low scores below the midpoint of 3.5 on the independent variables: FOTH (12.9%), FSELF (21.0%) and CANX (16.0%). Thus, these measures were slightly skewed to the right (−0.71 to −1.2). Therefore, this data set does not tell us much about those participants with low scores on BINT, FOTH, FSELF and CANX.

### 4.3. Contributions of the Study

Within the context of taking the first booster vaccine to address the continued COVID-19 pandemic, this research makes several contributions that could assist those creating public health campaigns involving vaccinations. First, fear should be more precisely defined to include types of fear other than FSELF. In this study, FOTH (i.e., fear of COVID-19 for friends and families) was distinguished from fear of COVID-19 for self (FSELF). FOTH was demonstrated to be somewhat more important than FSELF in impacting persons to develop a positive attitude toward getting the first booster vaccine, but FSELF makes its own unique contribution to the intent to take the first booster vaccine. So, the results indicate that more vaccination campaigns should appeal to the fear for others than they have in the past, while not neglecting the more traditional appeals to fear of the disease for potential harm it could do to oneself. These results do demonstrate that FSELF was an important influence on the intent to get the first booster vaccine. The direct impact of FSELF should not be minimized. Those who live alone may not be equally influenced daily by normative influences and might be more driven by FSELF. Not only do others fear for people around them, but they may also fear for themselves, because their presence enhances the lives of others. They do not want to risk the financial future of their dependents, nor risk not being around for important moments in the lives of their family members (i.e., children’s birthdays, family events, etc.).

Another important contribution of this study is providing evidence that anxiety about COVID-19 is a distinct concept—it is not synonymous with FOTH or FSELF. The mean level of CANX (4.81) was statistically higher than the mean level of FSELF (4.72). Yet, this study does not provide clarity about the exact role of anxiety in the process of leading persons to getting the first booster vaccine, since it was part of the indirect relationships with BINT in Figure 2. Scrima et al. [2] do provide evidence that heightened anxiety created by uncertain information can short-circuit the impact of fear on BINT. Also, anxiety may become more important for the adoption of new vaccines, as the COVID-19 pandemic shifts more to an endemic mode. Individuals have more information to process about the risks for the newest vaccines. Individuals know how their bodies reacted to prior COVID-19 vaccines. Overall, less uncertainty exists compared to the first COVID-19 vaccines.

Relevant to this discussion is Ohman’s [32] view of anxiety—that it often occurs when a person is not effectively able to resolve or fully cope with the threat being faced. Such could have been the case with the original COVID-19 vaccine and subsequent booster shots not eliminating the possibility of contracting the disease but reducing the likelihood of hospitalization or death. An unanswered issue is whether CANX resulted from the incomplete control of danger and fear of FOTH and FSELF.

Another possibility is that anxiety is more prevalent when the noxiousness of the threat is not intense enough to generate FOTH or FSELF. So et al. [29] found that anxiety was a better predictor than fear in their experimental study of the intention to adopt a meningitis vaccine. That study did not generate significant levels of fear of contracting meningitis. Perhaps the problem of so many carriers of COVID-19 being asymptomatic may have led to more anxiety, due to the great uncertainty of dealing with an unseen source, much like COVID-19 during the initial wave. Many questions remain about COVID-19 related anxiety and getting vaccines that this study cannot answer. The role of anxiety versus fear might be better answered by experimental studies, such as those conducted by So et al. [29] and Roberto and associates [16,18]. However, generating fear in experimental studies is difficult and carries with it many ethical issues. Studying naturally occurring events such as COVID-19 does not have this problem, but such severe diseases are uncommon.

### 4.4. Shifting Conditions for the Latest COVID-19 Vaccines

Approaches to FOTH, FSELF and CANX may need to shift to attract those in the USA to take the latest COVID-19 vaccine, as many facets of life in the USA recover. Although the number of deaths and hospitalizations are still high by traditional standards, the sense of urgency today is much diminished from the first week of December 2021, when the number of booster vaccines in arms exceeded a million vaccines on one day in the USA [41]. That was the period when data for this study were collected.

### 4.5. Limitations

#### 4.5.1. Lack of Connection to Message Design Strategy

A significant limitation of this study is the lack of direction for a message design strategy, as the COVID-19 pandemic moves into a more endemic mode, like the seasonal flu. According to the CDC [42,43], none of the COVID-19 booster vaccines have approached the vaccination rate for adults for influenza in the USA over the past few years. Future studies could compare individuals’ flu vaccine uptake compared to the COVID-19 vaccine uptake, to unearth similarities and contrasts.

#### 4.5.2. Limited Sample and Narrow Range of Participants

We recognize that the sample used in this study was far from adequate in many ways. The participants of this study were limited in age range and other demographics, due to using Amazon Mechanical Turk (AMT). Our sample was very educated and predominantly white, which is not uncommon on AMT. It was biased towards males (56%). The absence of very few participants aged 50 and over limited our consideration of the impact of age on the levels of FOTH, FSELF and CANX related to COVID-19. Unfortunately, those over 50 are more likely to suffer more severe effects from catching COVID-19. This also provides an opportunity to recognize that efforts made on vaccine uptake may need to be made based on different demographics, such as age.

#### 4.5.3. Problem of Political Division

The importance of controlling the influence of political affiliation is especially evident in the analysis in Figure 2. Strategically, this presents a problem for media strategy in regions that are politically more conservative. The presentation of FOTH for controversial vaccines like COVID-19 may necessitate different tactics.

### 4.6. Future Research

#### 4.6.1. Focus on Fear for Others and Fear for Self

Among the most vulnerable populations in the USA, implementing influence messages capitalizing on fear for others is an obvious extension of this study, while continuing to promote the need to protect oneself. Addressing these issues will help current and future issues when it comes to COVID-19 booster vaccines, as it seems COVID-19 will not be leaving in the near future.

#### 4.6.2. Salient Features of Effective Advertisements

One issue regards the source of the ads for vaccines. In fall 2023, major pharmaceutical companies initiated frequent television commercials focusing on the pandemic not being over. They promoted their new updated vaccine to address new variants. Television remains an effective means to get messages to more vulnerable populations during major sporting events. However, many questions remain without satisfactory answers. Are those over the age of 60 influenced by these commercials to take the new COVID-19 updated vaccines from the pharmaceutical companies? Are they more influenced by recent information from sources from the government without any profit motive, like the CDC? Increasingly, social media platforms are becoming more prevalent, but it is unclear how these new media may change the strategy.

Another consideration is the message characteristics of COVID-19 commercials. What is the appropriate frequency of the same commercial versus variations on the theme at the same rate? Is a mixture of messages from the drug manufacturers and government sources, like the CDC, desirable? Are charming narratives by well-known persons on commercials for drug manufacturers more influential than striking risk factors from the CDC? Do these messages in the media lead to further investigation with normative influence from one’s doctor or close friends? Should the intensity of the commercial provoke a fear response as high as during the peak of deaths during the pandemic? More difficult to answer is whether it is possible to foster some level of fear for others, fear for self, or related anxiety for those who are extremely hesitant in a very politically charged culture.

#### 4.6.3. Clarification of the Role of Anxiety

The research of Scrima et al. [2] implicates the role of false and problematic narratives in the generation of high levels of anxiety that may arise from various sources such as the media choices or discussions within their social networks. Those were not major influences in this study because the population taking COVID-19 booster vaccines already had taken the original vaccines, but these issues do need to be within the awareness of public health campaigns.

## 5. Conclusions

New ground was broken in this research that broadened the social dimensions of the construct of fear that is important to vaccination intentions. It was argued that fear of COVID-19 needed to be divided into the more discrete components of fear for others and fear for self. Fear for others was found to be a strong type of fear influencing the participants in their expressed intent to get the first booster shot. The argument made by So et al. [29], that fear and anxiety are separate emotional constructs with different action tendencies, was supported. Fear for others was particularly predictive of intent to take the first booster shot. The presence of indirect effects provide impetus for future studies to explore the inter-relationships of fear for others, fear for self and anxiety about whatever new vaccine is facing us. Taken together, the influences of fear for others, fear for self and anxiety about the disease are important to the message strategies for those developing and distributing vaccines to combat serious public health issues.

## Figures and Tables

**Figure 1 vaccines-12-00047-f001:**
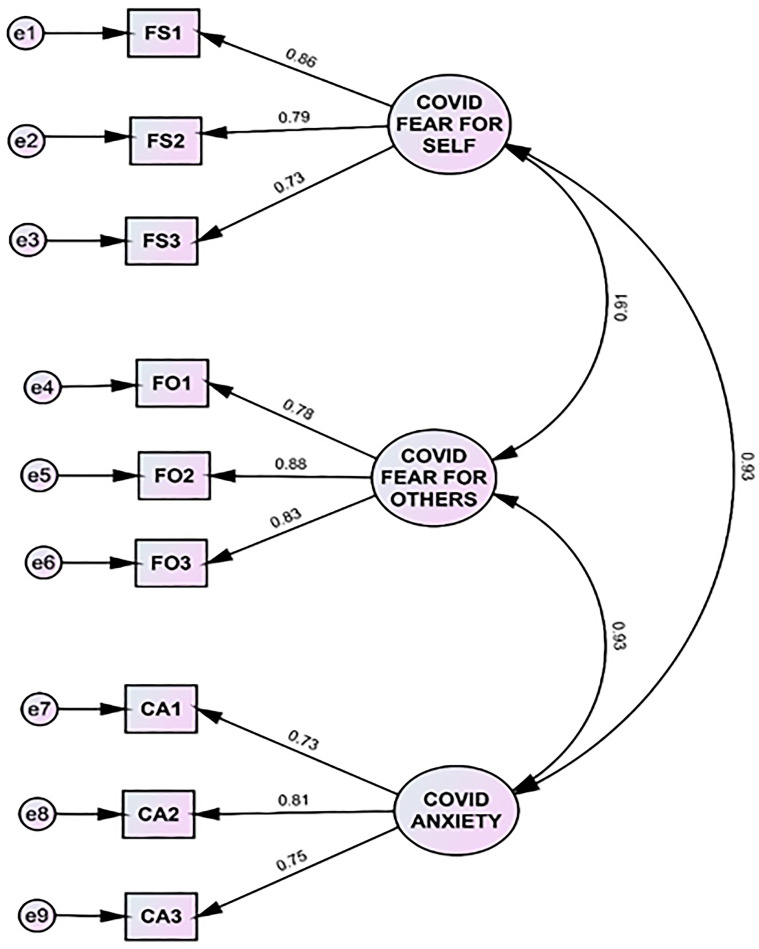
The confirmatory factor analysis of three COVID-19 related scales: Fear for Self (FSELF), Fear for Others (FOTH), and COVID-19 Anxiety (CANX) had a moderate fit.

**Figure 2 vaccines-12-00047-f002:**
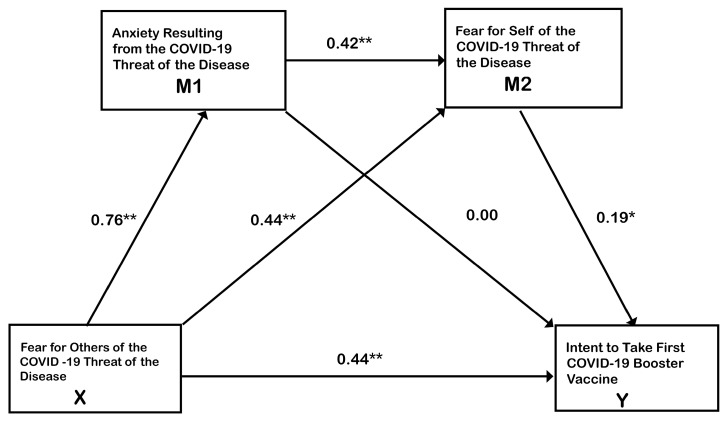
The Hayes (2022) PROCESS version 4.1 model 6 procedure indicated that Fear for Others (FOTH), and Fear for Self (FSELF) were significant direct predictors of the intent to take the first booster vaccine (BINT), *p* < 0.0001. The significant standardized indirect effect of X > M2 > Y was 0.08, *p* < 0.01. The standardized indirect effect of X > M1 > M2 > Y was 0.06, *p* < 0.01. COVID anxiety (CANX) was only a significant indirect predictor of BINT. ****** indicates *p* < 0.0001, * *p* < 0.002, N = 481.

**Table 1 vaccines-12-00047-t001:** Means, standard deviations, and Pearson correlations of study variables.

	M	SD	1	2	3	4	5	6	7
1. Fear for Others	5.09	1.40	1	0.78 **	0.78 **	0.14 **	−0.03	−0.11 *	0.63 **
2. Fear for Self	4.72	1.52	0.78 **	1	0.77 **	0.17 **	−0.12	−0.15 **	0.57 **
3. COVID Anxiety	4.81	1.42	0.78 **	0.77 **	1	0.13 **	−0.01	−0.11 *	0.52 **
4. Political Affiliation (Democrat)	0.61	0.49	0.14 **	0.17 **	0.13 **	1	−0.60 **	−0.57 **	0.24 **
5. Political Affiliation (Republican)	0.19	0.39	−0.03 *	−0.12	−0.01	−0.60 **	1	−0.22 **	−0.12 *
6. Political Affiliation (Independent)	0.17	0.38	−0.11 *	−0.15 **	−0.11 *	−0.57 **	−0.22 **	1	−0.14 **
7. Booster Indent	5.27	1.48	0.63 **	0.57 **	0.52 **	0.24 **	−0.12 *	−0.14 **	1

Note: Maximum political affiliation of dichotomous variables is 1. Dichotomous variables.are Spearman Correlations. N = 481. ** *p* < 0.01. * *p <* 0.05.

## Data Availability

Access to data for this study are in an Excel file along with an Excel file detailing the variables. SPSS resources also are located there. It can be accessed at: https://osf.io/42q3f/?view_only=fef66d50888642bb9da6f29b27cbb77d (accessed on 29 December 2023). Full data are contained at this site.

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
