# Peer review of "A Survey of Fear for Others, Fear for Self, and Pandemic Anxiety Predicting Intention to Take the First Booster Vaccine to Combat COVID-19"

_vaccines, 2023, doi:10.3390/vaccines12010047_

Round 1

Reviewer 1 Report

Comments and Suggestions for Authors

After reading the work, it seems the author was transferring work from a thesis into this manuscript. The first mistake is not to follow the instructions to authors and the second mistake is to have the writing poorly organized. The subsections seem to be a direct extract from another source.

Other comments include:

1. Introduction must be established from awareness and NOT records. The reveal of records are nothing less than support and therefore, these should be part of the discussion. Why is there need to carry out this work? Who are the target readers and what is the target field of study?

2. The structure of method and findings are mixed between psychology, decision science and theory of acceptance. It would be good to theme it on one ground and expand views from this direction.

3. I am not sure what could the findings be because both results and discussions are poorly organized.

4. Other comments in the attached file.

Author Response

We thank Reviewer 1 for perceptive comments. We can see why the reviewer
would raise these issues.

Reviewer 1 said, "The subsections seem to be a direct extract from another source."

The format of the abstract came from instructions from Vaccines that suggested these subsections. Perhaps we have misinterpreted those instructions.

In point 1, the reviewer said, " Why is there need to carry out this work? Who are the target readers and what is the target field of study?

We think that some of these issues now are covered more directly in the introduction. The audience is public health officials who design messages that encourage the uptake of COVID-19 booster vaccines. The focus is upon the use of appeals to fear for others, fear for self, and anxiety.

The reviewer is correct that fear appeals are covered from psychology, sociology, and communication. No one field covers influence. Moreover, appeals to fear for others have happened only rarely and not just in one place. It important to bring together the places where fear for others has been used as a motivation.

Reviewer 2 Report

Comments and Suggestions for Authors

The Scudder and DeBeck paper well describes how three emotions can predict the intention to undergo first booster vaccination against SARS CoV-2

The study is well organized, and three independent variables (fear for other vs fear for self vs pandemic anxiety) are correctly identified and studied. Results have shown that fear is a direct predictor of intention to vaccinate, and that anxiety is an independent variable of intention to receive the booster. Although further studies are necessary to better understand the nature of the indirect effects related to anxiety and fear in this behavioral and decision-making perspective.

Particular attention must be to citations in the text. Starting from reference 24 in the text there is a discrepancy with the citation reported in the "References" paragraph (e.g. 24 in the text corresponds to Chen and in "References" to Warr).

Author Response

We thank Reviewer 2 for the positive comments.  We liked your comments so much that we have taken your comment “Results have shown that fear is a direct predictor of intention to vaccinate, and that anxiety is an independent variable of intention to receive the booster.” To create a better introduction.

I don’t know what happened starting with reference 24. Actually, some references were missing. I went through and tried to make sure the cites lined up with the references, but I don’t know what happened. The references have all been redone now. Some were added and some were deleted.

Reviewer 3 Report

Comments and Suggestions for Authors

Tittle

At the end of the title specify the study type

Abstract

Methods: Add what type of the study implemented

Results: supplement the results with data as much as possible

No discussion in the abstract to be deleted

Conclusions: Be more precise and short

No references in the abstract, to be deleted

Keywords: could add hesitancy

Introduction

Define COVID-19 in the first place

Literature numbering and (?) or [?,? or ?;?]

Replace page number with references (p. 723)

Materials and Methods

Line 180-184 unnecessary paragraphs

Specify what type of study is this??

Could you perhaps shorten the introduction? It is rather long and confusing if you read it through.

Advice to add the study objective just before the methods

RQ3 Are significant indirect effects present among FOTH, FSELF or CANX as predic- 218

tors of BINT?

Comments: No text under this tittle

Participants

There were 481 participants after 223 data quality checks resulting in some participants not conforming to procedures (i.e. not 224 completing the survey, did not respond in English, etc.).

The sample size description need to be moved from the methods section to the results section.

Explain how you arrived at this sample size.

Comments on the Quality of English Language

Satisfactory

Author Response

Thank you for your comments. They have clearly changed how we think about the presentation of this manuscript from the beginning to the end.

Title

At the end of the title specify the study type

This has been done.

Abstract

Methods: Add what type of the study implemented

This has been done.

Results: supplement the results with data as much as possible

We have attempted to do this.

No discussion in the abstract to be deleted

Conclusions: Be more precise and short

We think we have sharpened and shortened some areas, but addressing concerns of other reviewers has extended some areas.

No references in the abstract, to be deleted

 Keywords: could add hesitancy

 We have added this term. Thank you for the suggestion.

Introduction

Define COVID-19 in the first place

We have done this.

Literature numbering and (?) or [?,? or ?;?]

We know this was problematic and have attempted to fix this.

Replace page number with references (p. 723)

We have worked on this, but we have not fully mastered this documentation format.

Materials and Methods

Line 180-184 unnecessary paragraphs

Specify what type of study is this??

We have now identified it as a survey method.

Could you perhaps shorten the introduction? It is rather long and confusing if you read it through.

We have shortened it some and tried to make it less confusing. The entire beginning was deleted and replaced.

Advice to add the study objective just before the methods

RQ3 Are significant indirect effects present among FOTH, FSELF or CANX as predic- 218

tors of BINT?

That is now more clearly discussed.

Comments: No text under this tittle

Participants

There were 481 participants after 223 data quality checks resulting in some participants not conforming to procedures (i.e. not 224 completing the survey, did not respond in English, etc.).

The sample size description need to be moved from the methods section to the results section.

Explain how you arrived at this sample size.

We have moved the participant section to the results. We think how we arrived at 481 participants is more clear. Thirty of the original surveys were incomplete or we believe were submitted by automatic bots..

Reviewer 4 Report

Comments and Suggestions for Authors

1. Please fix the minor grammatical and punctuation mark errors

2. Abstract and the rest: For every abbreviation used in the manuscript, it should be defined first before using the shortened term. an example would be the term COVID-19. Note that the abstract stands alone apart from the whole manuscript. So COVID-19 should be defined both in the abstract and introduction.

3. Abstract [methods] lacks several important principles, namely study design, study setting, sampling method, and data collection. Please provide.

4. Abstract should not place in any citations, therefore the order of references should be refreshed. Citation number 1 should begin in the introduction. If abstract provided citations, then please redirect the citation in the manuscript accordingly.

5. The introduction is has a narrow perspective by bringing up USA as a reference study to explain about COVID-19 vaccinations. To enlight and open up the horizon of the introduction, it is highly insisted to elaborate and cite the following study entitled:

"Global acceptance and hesitancy of COVID-19 vaccination: A narrative review".

6. Materials and method, the study variables and materials are clearly mentioned, however the very basics of methods section should provide the following points: study design, study setting, sampling method, and data collection.

7. The study serves as a very interesting matter with some level of potential to describe predicting intentions for booster vaccines.

Comments on the Quality of English Language

Overall it's fine, just several minor mistakes should be looked back.

Author Response

  1. Please fix the minor grammatical and punctuation mark errors

I have tried to do this. With so many changes in the document, I may have created a few more problems.
Sometimes seeing them is hard with the way it is now formatted.

  1. Abstract and the rest: For every abbreviation used in the manuscript, it should be defined first before using the shortened term. an example would be the term COVID-19. Note that the abstract stands alone apart from the whole manuscript. So COVID-19 should be defined both in the abstract and introduction.

Thanks for this suggestion. We have tried to fix this.

  1. Abstract [methods] lacks several important principles, namely study design, study setting, sampling method, and data collection. Please provide.

We have tried to fix many of these problems. Sampling is addressed as having limitations of not being a random sample. The sample does have an advantage of geographic diversity across the USA, but it is not random in terms of gender, age, and economic levels.

  1. Abstract should not place in any citations, therefore the order of references should be refreshed. Citation number 1 should begin in the introduction. If abstract provided citations, then please redirect the citation in the manuscript accordingly.

Citations were removed from the abstract and renumbered.

  1. The introduction is has a narrow perspective by bringing up USA as a reference study to explain about COVID-19 vaccinations. To enlight and open up the horizon of the introduction, it is highly insisted to elaborate and cite the following study entitled:

"Global acceptance and hesitancy of COVID-19 vaccination: A narrative review".

We have broadened the focus to be more global, but our sample still mostly comes from the USA. We did not have good access to a truly random global sample. We have consulted a number of resources on global acceptance of booster vaccines and the acceptance of booster vaccines has many confounding issues. We have done very poorly on booster vaccines in comparison to other countries like the UK.

  1. Materials and method, the study variables and materials are clearly mentioned, however the very basics of methods section should provide the following points: study design, study setting, sampling method, and data collection.

With the limitations mentioned above in point 6, we have addressed these issues more completely.

  1. The study serves as a very interesting matter with some level of potential to describe predicting intentions for booster vaccines.

Comments on the Quality of English Language

Overall it's fine, just several minor mistakes should be looked back.

We apologize that the original had more of these aspects not controlled better. I am still working on the references. This is not a reference format in which I am very familiar.

Reviewer 5 Report

Comments and Suggestions for Authors

The present manuscript reports about the individual emotions related to uptake of booster dose of COVID-19 vaccine. Authors have studied the role of major fears with respect vaccine uptake comprising fear for self, for others and pandemic anxiety and drawn the conclusion based on the interdependence and independent effect of these variables among a set of population.

Following are the comments after review of the paper;

1.      The participant size for the study is quite limited to draw out conclusions with respect to predicting intention.

2.      Kindly add exclusion criteria and elaborate inclusion criteria.

3.      The process of data collection has not been explained clearly. The method and contents of survey and how the participants responses were recorded has to be added in detail.

4.      The time duration for data collection has been mentioned as start of December 2021 to middle of January 2022. Does this limited time duration give a clear idea of generalized participant responses that are representative of overall population owing to limited sample size?

5.      Have the authors analyzed any data with respect to difference of intention between first wave of COVID-19 and second wave of COVID-19?

6.      Was there any effect on intention with respect to type of vaccine and its impact in the population. This needs to be taken as one of the covariates since media and general public response played major role in the same.

7.      Kindly check the values of Pearson correlation for Booster intent in Table 1. The coefficient is denoted as 100.

8.      The participant size is N=481. Why N=511 was used as data set for Pearson correlation in Table 1?

9.      How the Democrat political affiliation emerged out to be significant factor for booster intent in spite of less significant Pearson coefficient values. The covariate political affiliation needs to be incorporated for other parties as well since they make a part of total participants.

10.  Authors should have looked into the role of comorbidities as one of the covariates in the present study. The same has not been discussed.

Author Response

  1. The participant size for the study is quite limited to draw out conclusions with respect to predicting intention.

We acknowledge this in the manuscript under limitations. This study was self-funded except for the university covering publication costs. We think even a greater limitation is that the data are not a random sample across gender, age, location, education, and economic status. We have been exploring funding to conduct a more extensive study.

  1. Kindly add exclusion criteria and elaborate inclusion criteria.

I think this is now covered. Mainly persons were excluded because they had not had the original COVID-19 vaccine that was necessary to get the first booster vaccine. That is no longer the case with the current updated COVID-19 vaccine in the USA that is no longer called a booster. At least that was the case with the Pfizer update that I received..

  1. The process of data collection has not been explained clearly. The method and contents of survey and how the participants responses were recorded has to be added in detail.

I think the process of using Amazon Turk has now been described better. We also talk about exclusion of 30 responses because they were incomplete or failed to answer the challenge questions accurately because they were likely answered by bots and not have valid worker identification numbers. A few were duplicates.

  1. The time duration for data collection has been mentioned as start of December 2021 to middle of January 2022. Does this limited time duration give a clear idea of generalized participant responses that are representative of overall population owing to limited sample size?

Although the sample had broad distribution across the USA, It was not a truly random sample One problem is that those over 65 were eligible to take it before the general population. So, a criterion was that a participant could not already have received the booster vaccine. We clearly do not have enough persons over 50 in this sample. This is clearly a limitation that we have tried to address.

  1. Have the authors analyzed any data with respect to difference of intention between first wave of COVID-19 and second wave of COVID-19?

We mostly had interview and discussion qualitative data after the original vaccines were administered. Those interviews and discussions provided the foundation that our data collection was built upon in the
second wave.

We do now have unanalyzed follow-up data that has yet to be processed. Unfortunately, we have
a bit less than half of the original participants that also participated in the follow-up to see if they really did get the booster vaccine or not.

We have been trying to get funding for a much larger data collection.

  1. Was there any effect on intention with respect to type of vaccine and its impact in the population. This needs to be taken as one of the covariates since media and general public response played major role in the same.

This is now addressed briefly in the study There were no significant differences among persons in this study receiving the Pfizer, Modena, & J&J original vaccines in terms of willingness to get the first booster vaccine..Those getting the J&J one dose original vaccine only numbered about 60.

  1. Kindly check the values of Pearson correlation for Booster intent in Table 1. The coefficient is denoted as 100.

You are correct. We have fixed that, but also expanded the items in the table.

  1. The participant size is N=481. Why N=511 was used as data set for Pearson correlation in Table 1?

This was a mistake in the table. The 511 number was the number of raw submissions before checking for being complete and answering the challenge questions correctly. You have a sharp eye. I don’t think the other four reviewers mentioned that.

  1. How the Democrat political affiliation emerged out to be significant factor for booster intent in spite of less significant Pearson coefficient values. The covariate political affiliation needs to be incorporated for other parties as well since they make a part of total participants.

Given your comments, we have redone the analyses to include those identifying as Democrats, Republicans, or Independents. I think only 3-5 did not identify with any party. Three different variables were used and dummy coded. So, a person would be coded as 1 for being of that party and a 0 for no.

Only identifying as a Democrat significantly influenced whether they intended to get the booster vaccine in Figure 3. The impact of being a Democrat or not being a Democrat had a large beta of .35 as a covariate in the analysis. Being a Republican or Independent did not have a significant  impact as a covariate.

  1. Authors should have looked into the role of comorbidities as one of the covariates in the present study. The same has not been discussed.

We did not address comorbidities, but did address whether they had complications from the original COVID-19 vaccine and whether they had contracted COVID-19. We also inquired about maintaining regular visits to their health professionals and getting a regular flu vaccine. None of these variables had a significant impact on their intention to get the booster vaccine.

If they had comorbidities, they likely would not have even taken the original COVID-19. I know a few persons who did not take the original vaccine for that reason.

Round 2

Reviewer 3 Report

Comments and Suggestions for Authors

NIL

Comments on the Quality of English Language

NIL so far

Reviewer 5 Report

Comments and Suggestions for Authors

Thank you for the explanations provided against each raised comments.